# The allometric growth relationship between electricity consumption and economics in China

**Yi Qi●\*, Mengyuan Tang, Dongsheng Dang, Caijuan Qi**

State Grid Ningxia Electric Power Eco-Tech Research Institute, Yinchuan, China

\* qiyi_2022@163.com

## Abstract

Electric power is the basic industry of a modern country. The rapid development of the power industry has promoted economic development and social progress. With the establishment of the carbon neutrality target of carbon peak, China's power industry is also facing new situations and new challenges. This paper innovatively introduces the concept of allometric growth in biology, and uses the data of provincial electricity consumption and economic development from 1972 to 2017, to study the allometric growth relationship between electricity consumption and economics in China at multiple scales. Combined with the spatial autocorrelation analysis method, the temporal and spatial characteristics of allometric growth between electricity consumption and economic development were researched based on the analysis of spatial correlation of GDP electricity consumption intensity. The results show that: (1) The provincial GDP electricity consumption intensity shows an aggregate distribution, and the local aggregate pattern changed significantly from 1972 to 2017. (2) The High-High Cluster gradually disappeared, and the Low-Low Cluster moved southward, appearing in the middle and lower reaches of the Yangtze River. (3) The differential growth coefficient of electricity consumption and economic development increased initially and decreased from 1972 to 2017. (4) GDP would increase by 7.4%, 20.1%, and 15.6% in the simulation of electricity consumption, increasing 10% at the three stages of China's economic development. (5) The allometric growth coefficients of electricity consumption and economic development are very different in the three economic belts, eight economic zones, and provinces with different characteristics at the different stages of economic development. Formulating differentiated regulation policies for regional economic development and electricity consumption will be conducive to the coordinated development of electric power production and the national economy.

## 1. Introduction

Lots of energy economics research results have proved that electricity consumption is a vital factor of production in a country. No matter what causal relationship exists between them, electricity consumption will promote the sustainability of economic growth and ensure the

**Data Availability Statement:** All relevant data are within the manuscript and its Supporting information files.

**Funding:** The author(s) received no specific funding for this work.

**Competing interests:** The authors have declared that no competing interests exist.

continuity of national prosperity [1]. Over the past few decades, with the rapid socio-economic development, world energy demand and consumption continued to grow steadily [2]. The production and sufficient supply of electric power provide an essential guarantee for economic prosperity and social stability. The production and sufficient supply of electricity provide essential conditions for economic development, the improvement of people's living standards and social progress. The shortage of electric power supply will bring a tremendous negative influence on the development of the national economy and people's life [3]. To ensure an uninterrupted、reliable、safe and economic power supply [4], forecasting power consumption in advance is crucial to the planning, analysis and operation of power systems.

China's electricity consumption has doubled since 2010, with an average annual growth rate of 7.8%. Driven by the new development pattern of domestic and international double cycle mutual promotion, electricity consumption will continue to grow, which has become the general trend of the development of the times. At the same time, the government work report put forward the goal of GDP growth of about 5.5%. In this context, studying the spatial and temporal characteristics of the allometric growth of power consumption and economic development not only helps the government to formulate policies to ensure power supply demand, but also provides a reference for the long-term planning of power industry development.

At present, many scholars have studied the relationship between electricity consumption and economic development. Since the oil crisis in the 1970s, energy consumption, especially the relationship between power consumption and economical development, has been the focus of economists and policy analysts [5].

The domestic and foreign research literature on electricity consumption and economic development has been prosperous. Foreign scholars' research on electricity consumption and economic development mainly focuses on whether there is a long-term synergistic relationship between them. Kraft. J. (1978) creatively used the time series data from 1947 to 1974 to verify the one-way causal relationship between energy consumption and economic growth in the United States The cointegration theory has been widely applied to many countries and regions [1,6–11]. Many scholars believe there is a long-term cointegration relationship between electricity consumption and economic development. Still, there are three views on the causal relationship between electricity consumption and economic growth. According to statistics, 31.15% of the literature supports the neutral hypothesis (no causal relationship between electricity consumption and economic growth), 27.87% of the literature supports protection hypothesis (one-way causal relationship between economic development and electricity consumption), 22.95% of the literature supports growth hypothesis (one-way causal relationship between electricity consumption and economic growth), 18.03% of the literature supports feedback hypothesis (mutual causal relationship between electricity consumption and economic growth) [12]. Abdulrashid Rafindadi [1] studied Japan's economic growth and electricity consumption. The results showed that electricity consumption would increase by 0.5040% for every 1% increase in economic development in the long run. In the short term, for every 1% increase in economic growth, power consumption will increase by 0.5840%.

China's research on related issues started late, but with the rapid economic development, China is the world's second-largest economy and the highest power consumption country [13]. China's energy security and power shortage have attracted more and more attention from scholars at home and abroad [14–18]. Wu Yuming and Li Jianxia studied the relationship between provincial electricity consumption and economic growth in China. Using the geographically weighted regression model of spatial econometrics, it was found that there was an unbalanced linkage relationship and local characteristics between the two [19]. Pan Wei and other scholars use econometric geography methods to study the relationship between China's electricity consumption, economic growth and carbon dioxide emissions. Studies have shown

that there is a cointegration relationship between electricity consumption, economic growth and carbon dioxide emissions [20]. Zhou Haiyan et al. constructed a panel analysis model of the relationship between provincial energy consumption and economic growth from the perspective of heterogeneous energy consumption, and analyzed the problem of energy sacrificial growth based on China's economic impact [21]. Domestic scholars pay more attention to the empirical analysis of electricity consumption and economic growth. China's electricity consumption and economic growth are endogenous [3] and reciprocal causation [19,22]. The relationship between power consumption and economic growth in each province show a non-equilibrium linkage relationship and local characteristics. The power demand and economic growth in different regions is very complex [23–25]. With other factors unchanged, for every 1% increase in regional economic growth, regional electricity demand can increase by 0.38% [3].

In summary, the existing research has done in-depth research on electricity consumption and economic growth and their relationship. The specific contributions are as follows: (1) Most studies have conducted empirical research on the relationship between electricity consumption and economic growth in China from the perspective of time series. (2) The relationship between China's provincial electricity consumption and economic growth has unbalanced linkage and local characteristics. In view of this, the innovation of this paper lies in: (1) The allometric growth model represents the relationship between the structural characteristics and physiological attributes of organisms and their allometric growth in biology. (2) There is still a lack of research on the relationship between electricity consumption and economic development in China based on multi-scale. This paper studies from multi-scale and multi-level, and accurately formulates policy guidance for differentiated regional economic development and electricity consumption control.

China has a vast territory, with noticeable spatial differences in natural resource endowment, economic development foundation and industrial structure characteristics. The utilization rate of resources varies greatly, and the potential to meet the needs of economic development is different [26]. Because of the "West-Central-East" gradient model, the eastern provinces of China have better spatial, economic and technological advantages than the western provinces [27]. The imbalance in regional economic development is pronounced, and the electricity consumption levels are far from each other. In different stages of development, due to energy policy and economic policy, the internal dependence between electricity consumption and economic growth is different [28,29]. Therefore, the complex relationship between electricity consumption and economic development shows an obvious scale effect. Therefore, understanding the characteristics of different regions can effectively integrate and allocate resources such as talents, funds, materials, and information, thereby accelerating the coordinated economic development of underdeveloped regions [30]. In addition, China's clean energy resources are abundant, and in recent years, China's clean and low-carbon process has been accelerating. Many indicators such as hydropower, wind power, photovoltaic, and nuclear power installed capacity under construction have remained the world's first, and the world's largest clean power generation system has been built. It has become an important force to promote the development of global clean energy. At this stage, the advantages of energy consumption mode change are huge, and it will continue to develop rapidly in the future. However, up to now, the research on the relationship between electricity consumption and economic development in China based on multi-scale is still lacking. And the existing research results lack accurate guidance to formulate differentiated regional economic development and power consumption control policies. Based on this, this paper introduces the allometric growth model and combines with spatial econometric analysis method to analyze the spatial correlation characteristics of electricity consumption intensity of regional GDP. Based on

different stages of economic development, this paper analyzes the allometric growth relationship between electricity consumption and regional economic development and the internal relationship between electricity consumption and the industrial structure of provinces, three economic zones and eight economic zones, and reveals the temporal and spatial heterogeneity characteristics of the evolution law of electricity consumption and economic development, which is of great significance for correctly predicting the demand of electricity market, formulating differentiated regional electricity consumption regulation policies and ensuring the coordinated development of electricity production and national economy.

## 2.Data and methods

### 2.1 research method

**2.1.1 Electric—economic allometric growth model.** Allometric growth, first adopted by British biologist Hesterley, is a term used to describe the disproportionate growth relationship of organisms. It is mainly used for some laws in organic organisms, namely nonlinear quantitative relationships. Many scholars have applied it in recent years to economic geography, especially in the relationship between urban population and built-up areas [31]. This paper introduces the allometric growth model to analyze the allometric growth relationship between electricity consumption and economic development, which is generally expressed by power function:

$$Y = aX^b \tag{1}$$

After taking logarithms on both sides, the above equation becomes:

$$LnY = bLnX + Lna \tag{2}$$

In the formula, Y represents a gross domestic product (GDP) (billion), X represents electricity consumption (billion kWh), a is proportional coefficient, and b is the scale coefficient, also known as the allometric growth coefficient. b>1 indicates that the economic growth rate is greater than the growth rate of electricity consumption; b = 1 indicates that electricity consumption and economic development grow at an equal rate; b<1 indicates that the economic growth rate is smaller than the growth rate of electricity consumption.

**2.1.2 spatial correlation model.**

1. global spatial autocorrelation
   Global Moran's I can measure the relationship between attribute values of adjacent spatial distribution objects, and the similarity of the attribute values of spatial adjacency or adjacent area units. The value is between -1 and 1, the positive value indicates that there is a positive correlation between the attribute values of the space object, and the negative value indicates that there is a negative correlation between the attribute values of the space object. When the value is close to -1/(n-1), it indicates no spatial autocorrelation of the attribute value. The formula is [32,33]:

$$I = \frac{n\sum_i\sum_j W_{ij}(X_i - \overline{X})(X_j - \overline{X})}{\sum_i\sum_j W_{ij}(X_j - \overline{X})^2} \tag{3}$$

In the formula, n is the number of provinces in the study area, $X_i$ and $Y_i$ are attribute values for provinces (municipalities) i and j, $\overline{X}$ is the average of attribute values, $W_{ij}$ is a space weight matrix, space adjacent to 1, not adjacent to 0.

2. local spatial autocorrelation

In order to further reveal spatial heterogeneity, grasp the aggregation and differentiation characteristics of local spatial elements, the local Moran's I is adopted, the calculation formula is [33]:

$$I_i = Z_i \sum_{j \neq 1}^{n} W_{ij} Z_j \tag{4}$$

In the formula, $Z_i$ and $Z_j$ are the standardized values of provincial i and j attribute values, indicating the deviation degree between provincial attribute values and mean values.

$\sum_{j \neq 1}^{n} W_{ij} Z_j$ is the weighted average of attribute value deviation of adjacent regions.

### 2.1.3 Division of economic region and economic development stage.

1. Division of economic region

The economic zone and economic zone are divided according to the Chinese Geography Course edited by Wang Jingai [34]:Three major economic zones: Eastern region: Beijing, Tianjin, Hebei, Liaoning, Shanghai, Jiangsu, Zhejiang, Fujian, Shandong, Guangdong and Hainan provinces and cities; the middle zone of China: Shanxi, Jilin, Heilongjiang, Anhui, Jiangxi, Henan, Hubei and Hunan Provinces; western region: Chongqing, Sichuan, Guizhou, Yunnan, Tibet, Shaanxi, Gansu, Qinghai, Ningxia, Xinjiang, Guangxi and Inner Mongolia provinces, municipalities and autonomous regions.

China's eight economic regions: Northeast China includes Liaoning, Jilin and Heilongjiang provinces; the northern coastal areas include Beijing, Tianjin, Hebei and Shandong; the eastern coastal areas include Shanghai, Jiangsu and Zhejiang provinces; southern coastal areas include Fujian, Guangdong and Hainan provinces; The middle reaches of the Yellow River include Shaanxi, Shanxi, Henan and Inner Mongolia; the middle reaches of the Yangtze River include Hubei, Hunan, Jiangxi and Anhui provinces; southwest includes Yunnan, Guizhou, Sichuan, Chongqing and Guangxi provinces, cities and districts; The northwestern region includes Gansu, Qinghai, Ningxia, Tibet and Xinjiang provinces and districts.

2. Division of stages of economic development

According to the standard division model of Chenery [35], China's economic development process is divided into the following stages: 1972–1978, the planned economy stage before Reform and Opening Up; 1979–2003, the production stage of primary products after the reform and opening up, 2004–2017, the early and middle stages of industrialization.

## 2.2 Data source and processing

The research scope includes 31 provinces, autonomous regions and municipalities in mainland China (excluding Hong Kong, Macao and Taiwan). Data on electricity consumption in China and provinces from 1972 to 2013 are derived from the Compilation of statistical data of power industry; Data on gross domestic product (GDP) and added value of three industries in China and provinces (municipalities) from 1972 to 2008 are derived from the Compilation of Statistical Data for 60 Years of New China; and electricity consumption data for 2014–2017 and economic data for 2009–2017 are available from the website of the National Bureau of Statistics (http://www.stats.gov.cn/tjsj/ndsj/).

Chongqing established a municipality in 1997, with statistics on electricity consumption and economic data in 1997. To ensure consistency of data, part of the analysis will be merged

with Sichuan Province; Hainan Province evolved from the Hainan Special Economic Zone established in 1988, so its indicators were calculated from 1988.

## 3. Results

### 3.1 Spatial Correlation between electricity consumption and economic development

**3.1.1 Spatial distribution characteristics of power consumption intensity in GDP.** The electricity consumption intensity of regional GDP, namely the annual total electricity consumption ratio to regional GDP, can describe the overall utilization efficiency of electricity resources in the process of regional economic development to a certain extent. Drawing spatial pattern maps of power consumption intensity of GDP for 1972 and 2017 (Fig 1) by ArcGIS software.

From 1972 to 2017, the power consumption intensity of the whole society decreased from 3110.59 kWh/million GDP in 1972 to 2529.72 kWh/million GDP in 2017, and the power consumption intensity of provincial and municipal GDP also changed significantly (Fig 1).

In 1972, the high-value areas of electricity consumption intensity of economic development were mainly distributed in Gansu and Ningxia in the northeast and western regions and gradually decreased from the high-value areas to the southwest and southeast. The low-value center was in Tibet, and the sub-low-value center was in Fujian and Guangdong. During this period, China's industrial structure is dominated by the primary industry, and the energy consumption structure is dominated by non-renewable resource coal. Northeast China and Gansu are rich in natural resources, thus forming many resource-based cities that are highly dependent on resources, so many high-value areas of electricity intensity are formed centered on cities.

In 2017, the high-value area of electricity consumption intensity of economic development was transferred to the northwest region centered on Gansu Province, and the secondary high-value area was in the middle reaches of the Yellow River and southwest region; the low-value center is still in Tibet, the Yangtze River Basin and Northeast China formed two low-value

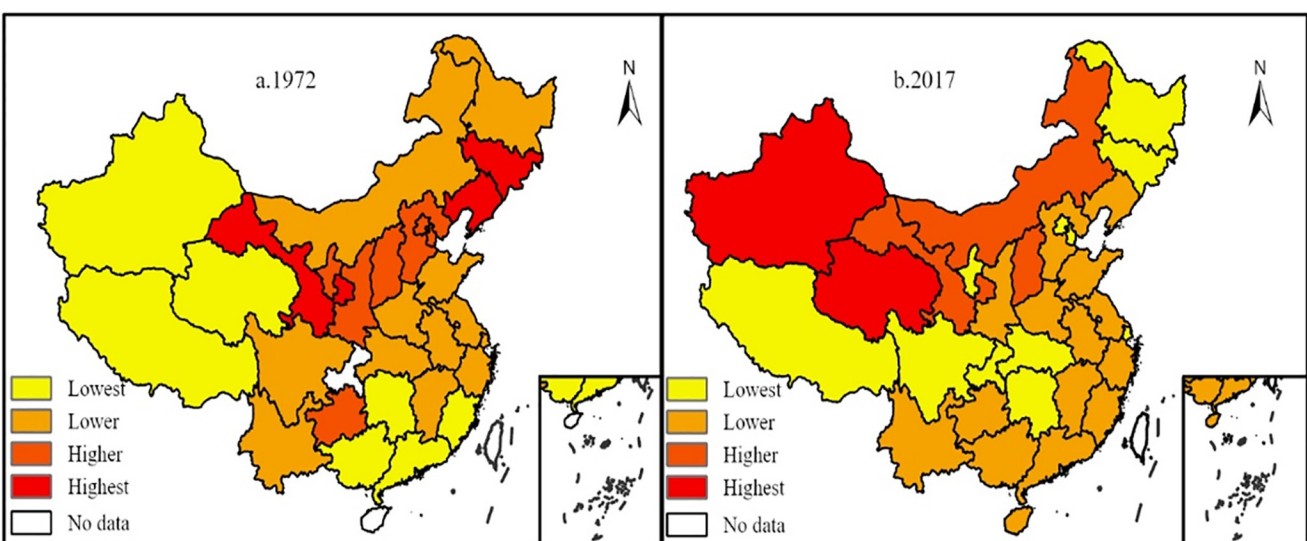

**Fig 1. Spatial distribution of power consumption intensity of GDP in provinces (cities).** Based on the standard map GS (2022) 4309 of the standard map service website of the Ministry of Natural Resources, the map boundary is not modified during ArcGIS production.

**Table 1. Moran's I statistics of GDP power consumption intensity.**

| particular year | 1972 | 1975 | 1985 | 1995 | 2005 | 2017 |
|---|---|---|---|---|---|---|
| Moran's $I$ | 0.2716 | 0.2092 | 0.2338 | 0.1876 | 0.1106 | 0.3007 |
| P | 0.01 | 0.03 | 0.02 | 0.03 | 0.06 | 0.01 |

zones (regions). It shows that as China's economic development enters a new stage, energy development also enters a new stage. The proportion of clean energy and the optimization of industrial internal structure have made outstanding contributions to the reduction of electricity intensity in the whole society. Gansu Province has a strong foundation of new energy, especially the rich wind and light energy resources in Gansu Province, thus forming a trend of electricity intensity growth centered on Gansu Province.

**3.1.2 global autocorrelation analysis.** Using Geoda software, according to the spatial distribution data of GDP electricity consumption intensity in urban areas of China in 1972, 1975, 1985, 1995, 2005, and 2017, the spatial global autocorrelation analysis of GDP electricity consumption intensity is carried out to reveal its spatial aggregation characteristics further. The statistical results of Moran's I are shown in It can be seen from Table 1 that, first of all, except that the Moran's I value in 2005 did not pass the test of 0.05 significance level, other years passed the test of less than 0.05 significance level, indicating that the electricity consumption intensity of GDP has a certain positive correlation in space, that is, the electricity consumption intensity of GDP is not completely random distribution in space, and there is a cluster distribution in some regions, which further confirms the existence of the aggregation state of the electricity consumption intensity of GDP in Fig 2. Secondly, the Moran's I values of cross-sectional data from 1972 to 2017 showed an increasing trend in volatility, indicating an enhanced aggregation distribution.

**3.1.3 local autocorrelation analysis.** The global Moran's I index can only reflect the overall spatial distribution pattern of the power consumption intensity of China's GDP, but cannot reflect the local visibility level and the spatial correlation pattern between adjacent regions. Therefore, it is necessary to further study the correlation between an eigenvalue in the local

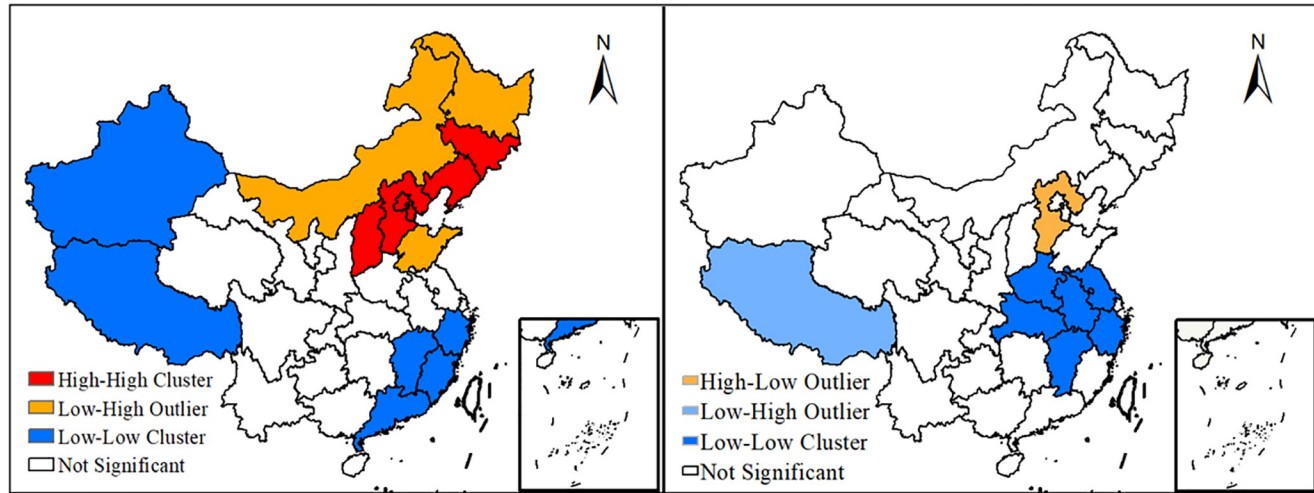

**Fig 2. Spatial differentiation characteristics of power consumption intensity of GDP.** Based on the standard map GS (2022) 4309 of the standard map service website of the Ministry of Natural Resources, the map boundary is not modified during ArcGIS production.

region and the eigenvalue in the adjacent local region. By analyzing the regional spatial correlation pattern of power consumption intensity of GDP in the study area, the LISA agglomeration map is drawn (Fig 2).

It can be seen from Fig 2 that the agglomeration pattern of electricity consumption intensity of GDP in each province and region has changed dramatically from 1972 to 2017. The type of High-High Cluster gradually disappeared, and Low-Low Cluster appeared in the Yangtze River Delta region, indicating that the spatial aggregation of power consumption intensity is weakening. In 1972, two high-high concentration areas centered on Liaoning Province and Hebei Province were formed, indicating that the economy of the northern region of China is more developed and is a "diffusion" region. The high-low concentration area centered on Sichuan Province is the economic center of the whole southwest. The low-high concentration area centered in Inner Mongolia is a "subsidence" area. Compared with North China, its economy is relatively backward [33]. In 2017, the most apparent spatial feature emergence of the Low-Low Cluster type in the Yangtze River Delta. The main reason for this phenomenon may be the rapid growth of economic development in this region and the higher value generated per unit of electricity consumption.

## 3.2 The allometric growth relationship between electricity consumption and economics

**3.2.1 China.** Drawing bilogarithmic coordinates of economic development (GDP) and electricity consumption (ELEC) for different periods 1972–1978, 1979–2003, 2004–2017 (Fig 3). The regression models of allometric growth in each period passed the test of 0.01 significance level.

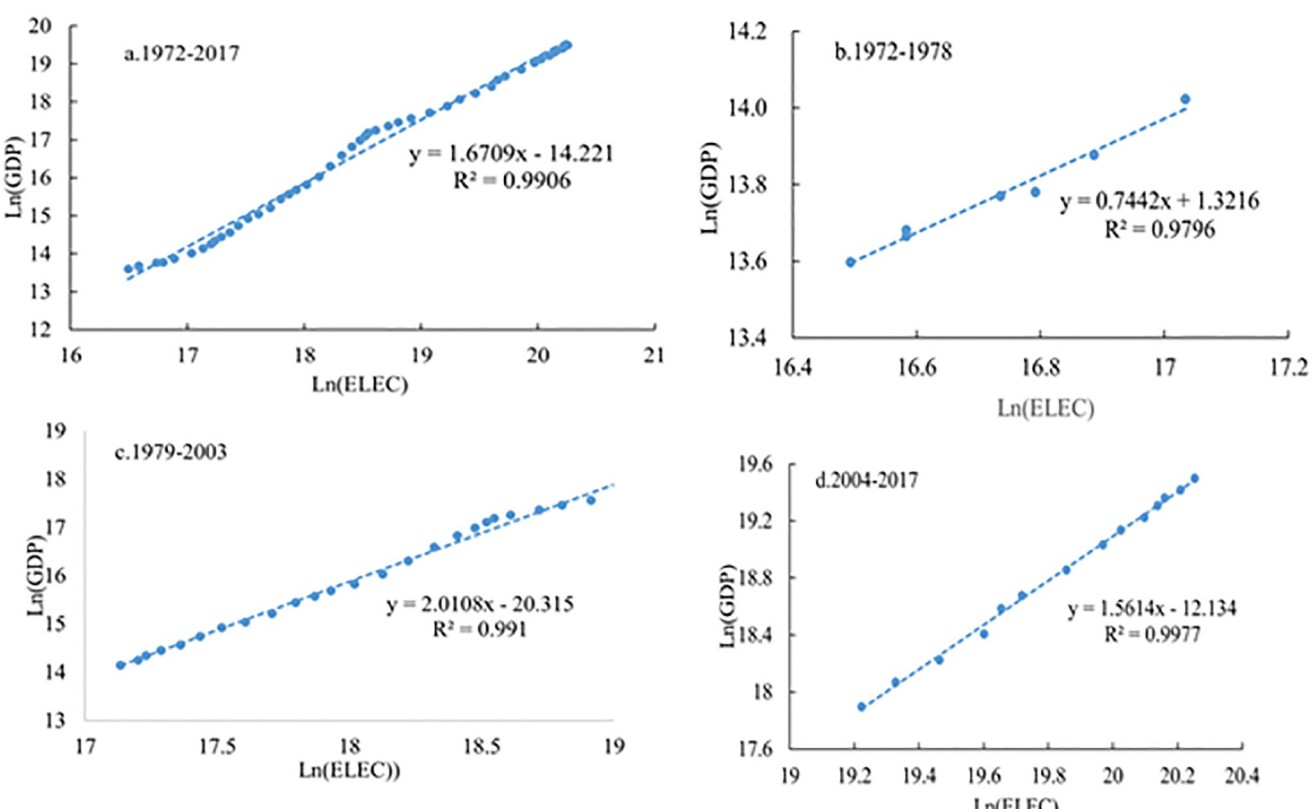

**Fig 3. Relationship between China's electricity consumption and economic development growth at different stages of economic development.**

As can be seen from Fig 3, there is a significant positive correlation between electricity consumption and GDP from 1972 to 2017. The scale coefficient of allometric growth is 1.6709. The economic growth rate is slightly larger than the growth rate of electricity consumption. The relative growth rate of economic development is 1.0978 times that of electricity consumption, but the growth process of the two fluctuates greatly. There are differences in the relationship between electricity consumption and economic growth in different stages of economic development. Before the Reform and Opening Up, the scale coefficient of allometric growth between 1972 and 1978 was the smallest, which was 0.7403, and its value was far less than 1, indicating that the GDP growth rate was far less than that of electricity consumption. After the reform and opening up, the scale coefficient increased rapidly to 2.0108 (R2 = 0.991) from 1979 to 2003, which was 2.70 times of that from 1972 to 1978. The growth rate of GDP was greater than that of electricity consumption. The scale factor for 2004–2017 fell to 1.5614, with GDP and electricity consumption close to constant growth. Therefore, against the background of the same increase of 10% in electricity resources, the economic growth rate was the smallest (7.4%) from 1972 to 1978 and the largest (10.01%) from 1979 to 2003.

**3.2.2 Three economic belts.** The electricity consumption and economic development of the three major economic zones have an abnormal growth relationship, and the temporal and spatial differences are obvious (Fig 4).

As a whole, from 1972 to 2017, the scale coefficient of abnormal growth of electricity consumption and economic development in the central region was the largest, while that in the

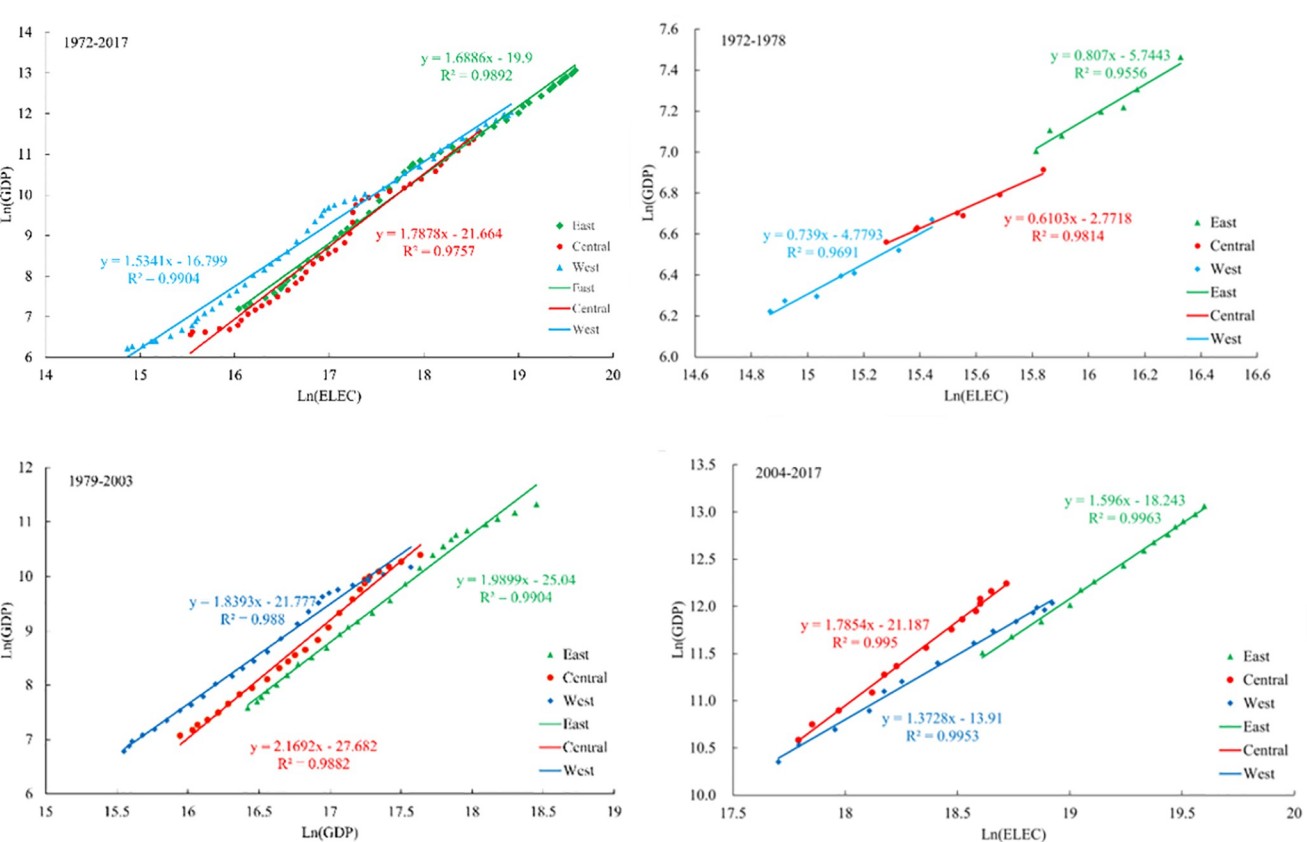

**Fig 4. Relationship between electricity consumption and GDP growth in the three economic zones.**

western region was the smallest, and there were differences in different stages of economic development.

From 1972 to 1978, the economic growth rates of the eastern, central and western regions were smaller than the growth rate of electricity consumption. From the perspective of scale coefficient, the order of abnormal speed degree from large to small was: central>eastern>western, indicating that under the background of increasing the same proportion of electricity consumption input in this period, the economic benefit of the eastern region was the largest, followed by the western region.

From 1979 to 2003, The allometric coefficients of electricity consumption and economic growth in the eastern and central and western regions are 1.9899 and 1.8393, respectively. With an increase of 10% in electricity consumption, the GDP of the eastern and central regions would increase by about 19%. During this period, the scale coefficient of economic growth in the central region exceeded 2, indicating that the speed of economic development was significantly higher than that of electricity consumption.

From 2004 to 2017, the scale coefficients of the abnormal growth of two of the three major economic zones have generally declined to a certain extent. Among them, the scale coefficient of the abnormal relationship between electricity consumption and regional GDP in the central region is the largest, which is 1.7854, indicating that the growth rate of regional GDP is greater than that of electricity consumption. The second is the eastern region, which is 1.596, and the smallest is 1.37 in the western region. The same increase of 10% in electricity resource input also increases the GDP of the eastern, central and western regions by 15.6%, 17.8%, and 13.1%, respectively.

**3.2.3 Eight economic zones.** SPSS correlation analysis shows that LnGDP and LnELEC in the eight economic zones at different stages of economic development are significantly positively correlated. Through the tests of 0.01 and 0.05 significance levels, they are in line with the law of allometric growth. The allometric growth scale coefficients of the two in different economic zones at different periods are shown in Table 2.

Throughout the whole development process, except for the continuous increase of the allometric growth scale coefficient in the southern coastal areas, the allometric growth scale coefficient in other economic areas has shown a trend of "increasing at first and then decreasing", among which the scale coefficient in northeast China is the largest in each stage of economic development. From 1972 to 1978, the scale coefficients of allometric growth in each economic zone were mostly less than 1. From 1979 to 2003, they were greater than 1. From 2004 to 2017, the allometric coefficients decreased. Northeastis still over 2, and the rest are between 1 and 2. This is related to the northeast revitalization strategy implemented in China since 2003 [36]. Due to the coordinated development of the region, the economic development of the northeast

**Table 2. Scale coefficients of electricity consumption and GDP abnormal growth at different stagesof economic development in the eight economic zones.**

| Regions | 1972–1978 | 1979–2003 | 2004–2017 |
|---|---|---|---|
| Northeast region | 1.0215 | 2.7375 | 2.1534 |
| Northern coast | 0.7148 | 2.128 | 1.5209 |
| Eastern coast | 0.8614 | 1.918 | 1.5649 |
| Southern coast | 0.3877 | 1.5045 | 1.6328 |
| Middle reaches of the Yellow River | 0.5685 | 1.9769 | 1.5343 |
| Middle reaches of Yangtze River | 0.5356 | 2.1687 | 1.67 |
| Southwest region | 0.8229 | 1.7864 | 1.6132 |
| Northwest region | 0.5839 | 1.8207 | 1.0687 |

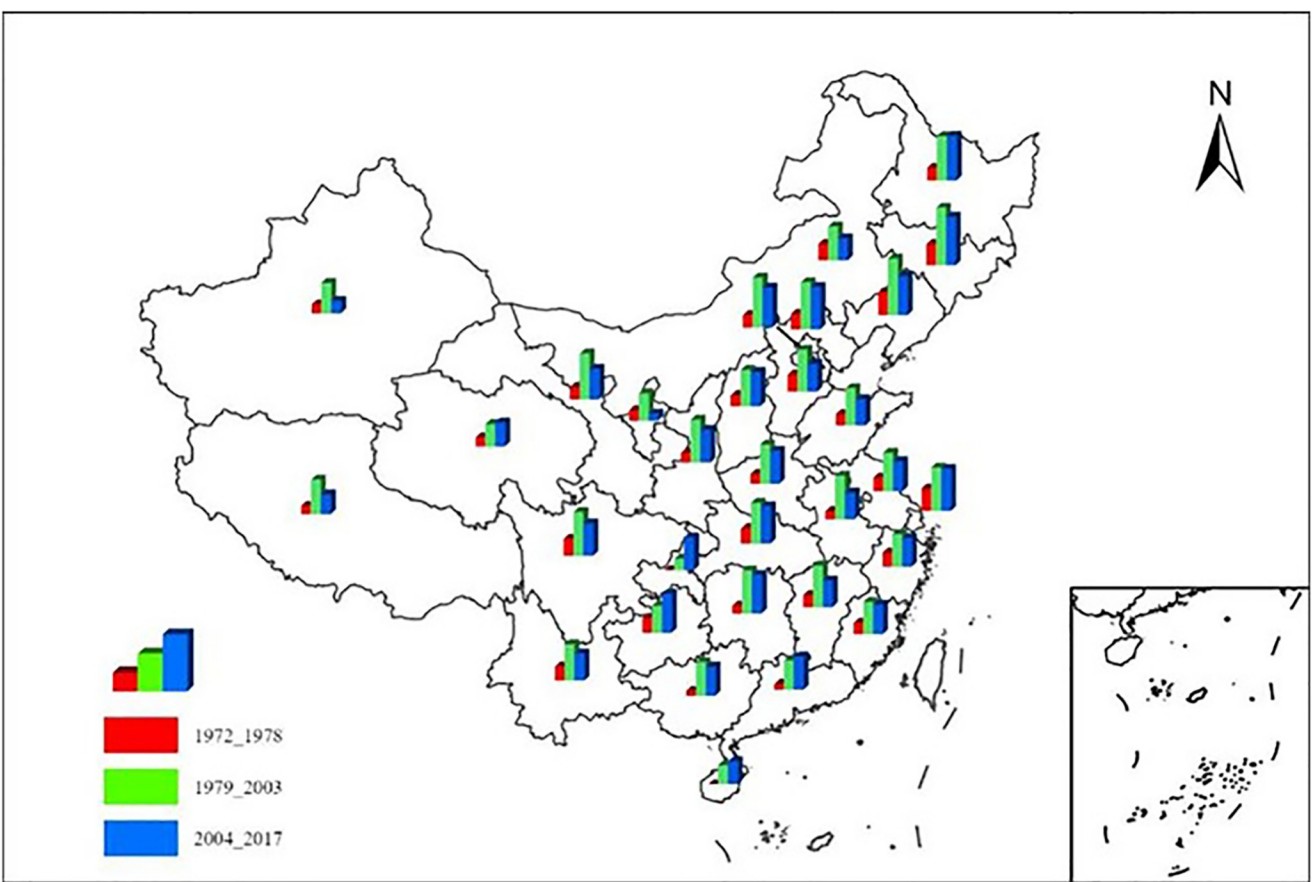

**Fig 5. Scale coefficients of abnormal growth of electricity consumption and regional GDP at different stages of economic development in each province (city) of China.** Based on the standard map GS (2022) 4309 of the standard map service website of the Ministry of Natural Resources, the map boundary is not modified during ArcGIS production.

region has achieved good results. The order of allometric growth scale coefficient from large to small from 2004 to 2017 is: Northeast China, Middle reaches of Yangtze River, Southern coast, Southwest region, Eastern coast, Middle reaches of the Yellow River, Northern coast, Northwest region. The input of power resources is also increased by 10%, and the GDP of Northeast China and the northwest region is increased by 21.5% and 10.6%, respectively, with a difference of more than two times. Under the influence of the northeast revitalization strategy and the new normal form of the economy, the industrial structure of the northeast region has been continuously adjusted. In 2015, the proportion of the tertiary industry in GDP exceeded that of the secondary industry, and the quality and efficiency of economic development have achieved certain results.

**3.2.4 Province (City).** Different provinces and autonomous regions in different stages of economic development of electricity consumption and GDP growth scale coefficient difference are prominent, the overall show a certain aggregation distribution pattern (Fig 5). Fig 5 is made by ArcGIS software. Generally, it can be divided into two categories. One is the J-type, and the essential feature is that the scale coefficient of allometric growth shows a continuous increase trend, mainly including Chongqing City, Guizhou, Guangdong, Hainan, Qinghai, and Heilongjiang. The other is mountain peak type, and the basic feature is that the scale

coefficient of allometric growth has experienced a process of increasing first and then decreasing. It mainly includes the remaining 25 provinces including Beijing, Tianjin, Tibet, etc.

From 1972 to 1978, the scale coefficients of allometric growth of provinces and cities are similar. Except that Liaoning Province (1.1557), Jilin Province (1.0678) and Shanghai City (1.1357) are greater than 1, the scale coefficients of allometric growth of other provinces and cities are less than 1. Generally speaking, the power consumption rate is higher than the economic development rate.

From 2004 to 2017, the allometric growth coefficient of each province showed obvious differentiation. The allometric coefficients of Jilin Province (2.4857), Heilongjiang Province (2.2925), Beijing City (2.1665), Shanghai City (2.1566) and Guizhou Province (2.0183) are greater than 2, which means that the economic development is obviously higher than the power consumption. The allometric coefficients of Ningxia and Xinjiang are all less than 1, which are 0.3573 and 0.6537 respectively, indicating that the economic development speed is lower than the electric power consumption. The number of allometric coefficients between 1 and 2 is 23, accounting for 76.7%, which shows that the economic development speed of these provinces is slightly higher than the power consumption. Among them, the allometric coefficient of Tibet is 1.0129, which indicates that the electricity consumption and regional GDP increase almost simultaneously.

## 4. Discussion

### 4.1 Thoughts on allometric growth

Allometric growth based on time and space is closely related to economic development and technological level. From the perspective of time, before 1978, due to the constraints of economic development level and technological development conditions, the allometric growth scale coefficient was less and the growth rate was slow. From 1979 to 2003, the rapid economic development and the relatively backward technological development level formed the extensive energy utilization mode, and the allometric growth scale coefficient growth rate was the largest at this stage. After 2004, science and technology and economy developed simultaneously, the concept of intensive development had a wide impact, and the growth rate of allometric growth scale coefficient slowed down. In this paper, the allometric growth relationship between electricity consumption and economics in China has been studied at multiple scales. In contrast, some articles from the perspective of water resources improved the value accounting method for water resources by assessing water efficiency and quality [37]. From a spatial point of view, the level of economic development in Northeast China is higher than that of electricity consumption, while the level of economic development in the central and western regions is lower than that of electricity consumption, indicating that the degree of energy intensive use in Northeast China is lower than that in the central and western regions. Therefore, with the progress of economic level and the continuous innovation of science and technology, the development of new energy has a significant impact on allometric growth. New energy has a good development prospect for optimizing the power structure and promoting the transformation of the power industry.

### 4.2 countermeasures and suggestions

1. Adjust power source structure.
   Since entering the new normal, China's economic growth rate has continued to slow down with double-digit high growth, and the economic growth rate has decreased, so that the growth rate of electricity demand has also decreased. With the increasingly prominent

environmental problems, China began to implement energy conservation and emission reduction policies in 2007. On the one hand, economic development needs the support of electricity. On the other hand, China's power industry is also a key area of pollutant emission reduction. Therefore, the power industry should gradually shift from meeting the "quantity" demand to focusing on meeting the "quality" demand. China's electric power production structure is quite different from that of other developed countries. Power production is dominated by thermal power. In 2017, thermal power accounted for 71.8% of all power generation, hydropower accounted for 18.3%, and non-water renewable energy generation accounted for only 6.0%. To solve the environmental bottleneck faced by China's power development and provide high-quality power energy for economic development, it is necessary to optimize the power supply structure. Changing the dominant position of non-clean energy in the energy structure and gradually increasing the proportion of clean energy can not only meet the demand of economic development for electricity, but also reduce the environmental pollution caused by thermal power coal, so as to achieve the purpose of emission reduction.

2. Optimize power layout.

China's eastern, central and western regions uneven distribution of energy resources. The power generation resources in the eastern and central regions are relatively scarce, but the power load is large. The centralized construction of a large number of coal-fired power plants has caused problems such as environmental capacity shortage in the eastern and central regions, excessive dependence on coal transportation for energy transportation, and the phenomenon of power shortage and coal shortage has repeatedly occurred. On the contrary, the western region is rich in power generation resources such as coal, hydropower and wind energy. However, due to the lagging economic development, the electricity load is relatively small and the electricity supply is relatively surplus. In recent years, clean energy such as wind power and photoelectricity has been greatly developed in Gansu, Xinjiang, and other places. However, due to the weak ability of local power grid to accept wind power, the limited local consumption ability and the insufficient transmission channel, the abandonment of wind and light is serious. Therefore, optimizing the distribution of power, speeding up the construction of power grid, through the introduction of clean energy replacement and transmission to outside the area of consumption, can achieve the effect of optimal allocation of resources and macro energy saving and emission reduction at the national level or in a large range, that is, to achieve the goal of reducing energy consumption in the transmission area and to achieve the purpose of pollution reduction in the receiving area. At the same time, wind power transmission can transform the resource advantages of Northwest China into economic advantages, alleviate the pressure of supply and demand and environmental protection in the Middle East, which is the objective requirement of implementing the national energy strategy and promoting the economic development of Northwest China.

3. Formulate differentiated regional economic development policies.

At present, there is still a large gap in the level and mode of economic development between the eastern, central and western parts of China, which also leads to significant differences in the relationship between electricity consumption and abnormal growth of regional GDP. Therefore, in the formulation of regional development policies, regional differences should be fully considered and adapted to local conditions. For the eastern region, we should develop in transition, attach importance to innovation-driven, develop new economic growth models such as sharing economy and platform economy, and reduce the pressure on resources, environment and electricity. For the western region, taking into account the

obvious differences in power consumption intensity between the eastern region and the western region, there is a large space to improve the efficiency of power input. It is necessary to transform in development, pay attention to technological progress and improve the efficiency of resource allocation. On the one hand, we should fundamentally change the current economic development model, curb the blind investment and low-level redundant construction of traditional industries with high energy consumption and high pollution, actively promote the development of emerging service industries and advanced manufacturing industries based on the direction of industrial structure diversification, gradually reduce the proportion of the secondary industry in the three industries, and gradually increase the proportion of the tertiary industry in the industrial composition, thereby reducing the energy consumption of economic development. On the other hand, China should strengthen the research and development of energy storage technology and formulate corresponding measures to vigorously promote the use of clean energy to ensure that the western region can obtain higher income from power output and realize the fairness of the impact of power development on regional economy.

## 5. Conclusions

Electricity development is closely linked to economic and social development. In this article, combined with the spatial autocorrelation analysis method, the temporal and spatial characteristics of allometric growth between electricity consumption and economic development were researched based on the analysis of the spatial correlation of GDP electricity consumption intensity. The conclusions are as follows:

1. From the perspective of the spatial distribution of electricity consumption intensity in GDP: In 1972, the high value areas of economic development electricity intensity were mainly distributed in Gansu and Ningxia in the northeast and western regions, and gradually decreased from the high value areas to the southwest and southeast. The low value center is in Tibet, and the second low value center is in Fujian and Guangdong. In 2017, the high value area of electricity intensity for economic development was transferred to the northwest region centered on Gansu Province, and the secondary high value area was in the middle reaches of the Yellow River and the southwest region; the low value center is still in Tibet, and the Yangtze River Basin and the northeast form two low value areas; from the perspective of spatial agglomeration characteristics: GDP and electricity intensity are not completely randomly distributed in space, but there is a certain spatial agglomeration relationship. From 1972 to 2017, the agglomeration pattern of electricity intensity in GDP of each province has undergone tremendous changes. The high-high star cluster gradually disappeared, and the low-low star cluster moved southward and appeared in the middle and lower reaches of the Yangtze River.

2. From 1972 to 2017, there was a significant positive correlation between China's electricity consumption and GDP. The proportion factor of China's electricity consumption and allometric growth in economic development increased first and then decreased. With the same 10% increase in electricity consumption, GDP will increase by 7.4%, 20.1%, and 15.6% in the three stages of economic development.

3. From the relationship between the three major economic belts and the eight economic zones and the allometric economic growth, there are obvious differences in the relationship between power consumption and GDP growth in the three major economic zones. From 1972 to 1978, the scaling coefficients of allometric growth in the three economic regions

were mostly less than 1, and the most obvious relationship between allometric growth was in the central region. From 1979 to 2003, the scale coefficient of allometric growth in the eastern, central and western economic regions reached the maximum. From 2004 to 2017, the power consumption and economic development scale coefficients of the eastern, central and western economic belts decreased to 1.596, 1.7854, and 1.37, respectively, which also increased by 10%. The GDP of the eastern, central and western economic belts increased by 15.96%, 17.85% and 13.7% respectively. There is a significant spatial difference between the scale coefficient of electric power consumption and economic development in the eight major economic regions and provinces, showing different characteristics in different stages of economic development.

4. From the perspective of the relationship between power consumption and allometric economic growth in provinces (municipalities): Different provinces and autonomous regions had significant differences in the scale coefficient of electricity consumption and GDP growth at different stages of economic development, showing a certain aggregation distribution pattern on the whole. From 1972 to 1978, the scale coefficient of allometric growth was similar among provinces and cities, and from 2004 to 2017, the allometric growth coefficient was significantly different among provinces and cities.

Based on the previous research results, this paper innovatively introduces the allometric growth model in biology, broadens the application scope of the method, and realizes the interdisciplinary application of the method. From the perspective of multi-scale and multi-level, this paper analyzes the relationship between electricity consumption and allometric economic growth, which can better provide theoretical support for the rational use of regional electricity in China, provide scientific basis for the formulation of differentiated regional electricity regulation policies, provide application value for cross-section empirical research, and promote the coordinated development of national economy and electricity production.

## Supporting information

**S1 Data.**
(XLSX)

## Author Contributions

**Conceptualization:** Yi Qi, Mengyuan Tang.

**Formal analysis:** Yi Qi.

**Validation:** Caijuan Qi.

**Writing – original draft:** Dongsheng Dang.

**Writing – review & editing:** Yi Qi.

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
