## [Decision Letter · Decision Letter 0]

9 Mar 2023

PONE-D-23-00464The Allometric Growth Relationship Between Electricity Consumption and Economic： A case of ChinaPLOS ONE

Dear Dr. Qi,

Thank you for submitting your manuscript to PLOS ONE. After careful consideration, we feel that it has merit but does not fully meet PLOS ONE’s publication criteria as it currently stands. Therefore, we invite you to submit a revised version of the manuscript that addresses the points raised during the review process.

Please reply to reviewers' comments carefully and make changes accordingly, including the literature review, method and specific suggestions for results. 

We look forward to receiving your revised manuscript.

Kind regards,

Hao Xue

Academic Editor

PLOS ONE

Journal Requirements:

3. We note that Figures 1,2 and 5  in your submission contain [map/satellite] images which may be copyrighted. All PLOS content is published under the Creative Commons Attribution License (CC BY 4.0), which means that the manuscript, images, and Supporting Information files will be freely available online, and any third party is permitted to access, download, copy, distribute, and use these materials in any way, even commercially, with proper attribution. For these reasons, we cannot publish previously copyrighted maps or satellite images created using proprietary data, such as Google software (Google Maps, Street View, and Earth). For more information, see our copyright guidelines: http://journals.plos.org/plosone/s/licenses-and-copyright.

      1. You may seek permission from the original copyright holder of Figures 1,2 and 5 to publish the content specifically under the CC BY 4.0 license. 

Additional Editor Comments:

Please reply to reviewers' comments and make changes accordingly.

Reviewers' comments:

Reviewer's Responses to Questions

**Comments to the Author**

1. Is the manuscript technically sound, and do the data support the conclusions?

Reviewer #1: Yes

Reviewer #2: Yes

2. Has the statistical analysis been performed appropriately and rigorously? 

Reviewer #1: Yes

Reviewer #2: Yes

3. Have the authors made all data underlying the findings in their manuscript fully available?

Reviewer #1: Yes

Reviewer #2: Yes

4. Is the manuscript presented in an intelligible fashion and written in standard English?

Reviewer #1: Yes

Reviewer #2: Yes

5. Review Comments to the Author

Reviewer #1: It is easy to understand this topic and the results while looking back at history. The economic development of China depends on our policies at different stages focusing on different regions and different industries. Data processing, visualization and statements are normal to varify the development facts of China, which can be a good reference for related studies. There are two points need improvement: accurate description all the figures like Line185-197, countermeasures thinking about our historic policies and advantage theories.

Reviewer #2: 1. The innovation of this paper needs to be highlighted in the abstract.

2. The author introduced the research background of this paper too much, but they does not explain the realistic background of this research very well

3. The literature review is not enough, the innovation of this paper and the contribution made by previous studies have not been clearly expressed.

4. The presentation of the method is very imprecise

5. Interpretation of results does not highlight important issues studied in this paper

6. The discussion section does not analyze the results of this paper, and provides enlightening thinking on related issues.

7. Compared with the available literature, what are the theoretical contributions and application values of this study? It is suggested to enhance the corresponding discussions in the conclusion part

8. English presentation requires more refinement

9. It is suggested that the authors add more descriptions of the reasons for choosing China as a case study, the following literature should be helpful for your research, the following literature should be helpful for your research：(1) Decoupling economic growth from water consumption in the Yangtze River Economic Belt, China. (2)Coordination of the Industrial-Ecological Economy in the Yangtze River Economic Belt, China. (3) Development of multidimensional water poverty in the Yangtze River Economic Belt, China.

6. PLOS authors have the option to publish the peer review history of their article (what does this mean?). If published, this will include your full peer review and any attached files.

Reviewer #1: No

Reviewer #2: No

While revising your submission, please upload your figure files to the Preflight Analysis and Conversion Engine (PACE) digital diagnostic tool, https://pacev2.apexcovantage.com/. PACE helps ensure that figures meet PLOS requirements. To use PACE, you must first register as a user. Registration is free. Then, login and navigate to the UPLOAD tab, where you will find detailed instructions on how to use the tool. If you encounter any issues or have any questions when using PACE, please email PLOS at figures@plos.org. Please note that Supporting Information files do not need this step.<quillbot-extension-portal></quillbot-extension-portal>

---

## [Author Response · Author response to Decision Letter 0]

5 Jun 2023

Dear editors, peer reviewers : 

Thank you very much for your comments on our manuscript. These views have certain reference value for the revision and improvement of this article. We have carefully made a detailed answer to the opinions of the two reviewers, and made detailed amendments to the paper according to the opinions of the reviewers. In order to facilitate your review, the modified part is now annotated in blue font. The replies and amendments to your manuscript are as follows :

Additional requirements：

The figures 1,2 and 5 in this paper are based on the standard map GS ( 2022 ) 4309 of the standard map service website of the Ministry of Natural Resources of China. The map boundary is not modified and there is no copyright dispute, which meets the requirements of your copyright guide.

一、 Opinions of Reviewer #1:

Recommendation 1 : Accurately describe all the numbers of lines 185-197, and combine China 's historical policy and advantage theory to think about countermeasures.

Response : Thank you for your valuable comments on our manuscript. We analyzed the data according to your opinion. Specific modifications are as follows:

From 1972 to 2017, the power consumption intensity of the whole society decreased from 3110.59 kWh/million GDP in 1972 to 2529.72 kWh/million GDP in 2017, and the power consumption intensity of provincial and municipal GDP also changed significantly (Figure 1). In 1972, the high-value areas of electricity consumption intensity of economic development were mainly distributed in Gansu and Ningxia in the northeast and western regions and gradually decreased from the high-value areas to the southwest and southeast. The low-value center was in Tibet, and the sub-low-value center was in Fujian and Guangdong. During this period, China 's industrial structure is dominated by the primary industry, and the energy consumption structure is dominated by non-renewable resource coal. Northeast China and Gansu are rich in natural resources, thus forming many resource-based cities that are highly dependent on resources, so many high-value areas of electricity intensity are formed centered on cities.(line244-257) 

It shows that as China 's economic development enters a new stage, energy development also enters a new stage. The proportion of clean energy and the optimization of industrial internal structure have made outstanding contributions to the reduction of electricity intensity in the whole society. Gansu Province has a strong foundation of new energy, especially the rich wind and light energy resources in Gansu Province, thus forming a trend of electricity intensity growth centered on Gansu Province. (line261-267) 

二、 Opinions of Reviewer #2:

Recommendation 1 : The innovation of this paper needs to be highlighted in the abstract.

Response : Thank you for your valuable comments on our manuscript. Based on your comments, the specific modifications we have made are as follows:

Electric power is the basic industry of a modern country. The rapid development of the power industry has promoted economic development and social progress. With the establishment of the carbon neutrality target of carbon peak, China 's power industry is also facing new situations and new challenges. This paper innovatively introduces the concept of allometric growth in biology. This paper innovatively introduces the concept of allometric growth in biology, and uses the data of electricity consumption and economic development in the province from 1972 to 2017 to study the relationship between electricity consumption and allometric economic growth in China at multiple scales. Combined with the spatial autocorrelation analysis method, based on the spatial correlation analysis of GDP electricity intensity, the spatial and temporal characteristics of allometric growth of electricity consumption and economic development are studied. (line6-16)

Recommendation 2 : There are too many research backgrounds introduced, but the realistic background of this study is not well explained.

Response : Thank you very much for your suggestion. According to your suggestions, we have revised the introduction part and strengthened the content of the realistic background part of this study. Specific modifications are as follows:

China 's electricity consumption has doubled since 2010, with an average annual growth rate of 7.8 %.Driven by the new development pattern of domestic and international double cycle mutual promotion, electricity consumption will continue to grow, which has become the general trend of the development of the times. At the same time, the government work report put forward the goal of GDP growth of about 5.5 %. In this context, studying the spatial and temporal characteristics of the allometric growth of power consumption and economic development not only helps the government to formulate policies to ensure power supply demand, but also provides a reference for the long-term planning of power industry development. (line47-54)

Recommendation 3 : The literature review is not enough, the innovation of this paper and the contribution of previous studies have not been clearly expressed.

Response : Thank you very much for your valuable comments. According to your suggestions, first of all, it enriches the content of the literature review, and re-condenses the contributions made by the predecessors and the innovation of this paper. Specific adjustments are as follows

Wu Yuming and Li Jianxia studied the relationship between provincial electricity consumption and economic growth in China.Using the geographically weighted regression model of spatial econometrics, it was found that there was an unbalanced linkage relationship and local characteristics between the two[31]. Pan Wei and other scholars use econometric geography methods to study the relationship between China 's electricity consumption, economic growth and carbon dioxide emissions. Studies have shown that there is a cointegration relationship between electricity consumption, economic growth and carbon dioxide emissions[35]. Zhou Haiyan et al.constructed a panel analysis model of the relationship between provincial energy consumption and economic growth from the perspective of heterogeneous energy consumption, and analyzed the problem of energy sacrificial growth based on China 's economic impact[36]. Domestic scholars pay more attention to the empirical analysis of electricity consumption and economic growth. (line83-97)

In summary, the existing research has done in-depth research on electricity consumption and economic growth and their relationship. The specific contributions are as follows : (1) Most studies have conducted empirical research on the relationship between electricity consumption and economic growth in China from the perspective of time series. (2) The relationship between China 's provincial electricity consumption and economic growth has unbalanced linkage and local characteristics. In view of this, the innovation of this paper lies in : (1) The allometric growth model represents the relationship between the structural characteristics and physiological attributes of organisms and their allometric growth in biology. (2) There is still a lack of research on the relationship between electricity consumption and economic development in China based on multi-scale. This paper studies from multi-scale and multi-level, and accurately formulates policy guidance for differentiated regional economic development and electricity consumption control. (line104-116)

Recommendation 4 : Inaccurate presentation of methodology

Response : Thank you very much for your valuable suggestions. Our description of the method adds the following :

Global Moran I can measure the relationship between the attribute values of adjacent spatial distribution objects, and the similarity of the attribute values of spatial adjacency or adjacent area units. (line188-190)

In order to further reveal the local instability and spatial heterogeneity, grasp the aggregation and differentiation characteristics of local spatial elements. (line200-201)

Recommendation 5 : The explanation of the results does not highlight the important issues studied in this paper.

Response : Thank you for your valuable comments. Based on your suggestions, we further explained the results by adding the following :

During this period, China 's industrial structure is dominated by the primary industry, and the energy consumption structure is dominated by non-renewable resource coal. Northeast China and Gansu are rich in natural resources, thus forming many resource-based cities that are highly dependent on resources, so many high-value areas of electricity intensity are formed centered on cities. (line252-257)

It shows that as China 's economic development enters a new stage, energy development also enters a new stage. The proportion of clean energy and the optimization of industrial internal structure have made outstanding contributions to the reduction of electricity intensity in the whole society. Gansu Province has a strong foundation of new energy, especially the rich wind and light energy resources in Gansu Province, thus forming a trend of electricity intensity growth centered on Gansu Province. (line262-268)

This is related to the northeast revitalization strategy implemented in China since 2003.Due to the coordinated development of the region, the economic development of the northeast region has achieved good results. (line462-464)

Under the influence of the northeast revitalization strategy and the new normal form of the economy, the industrial structure of the northeast region has been continuously adjusted. In 2015, the proportion of the tertiary industry in GDP exceeded that of the secondary industry, and the quality and efficiency of economic development have achieved certain results. (line469-473)

Recommendation 6 : The discussion section does not analyze the results of this article and provide enlightening thinking on related issues.

Response : Thank you for your valuable comments. We adopted your suggestion. In order to strengthen the discussion of the results, this paper will discuss the discussion as the fourth part of the article in the process of modification. 4.2 is based on the thinking of related issues put forward relevant policy recommendations. In the results section, the discussion of related content is strengthened. The specific modifications are as follows :

4.1 Thoughts on allometric growth

Allometric growth based on time and space is closely related to economic development and technological level. From the perspective of time, before 1978, due to the constraints of economic development level and technological development conditions, the allometric growth scale coefficient was less and the growth rate was slow. From 1979 to 2003, the rapid economic development and the relatively backward technological development level formed the extensive energy utilization mode, and the allometric growth scale coefficient growth rate was the largest at this stage. After 2004, science and technology and economy developed simultaneously, the concept of intensive development had a wide impact, and the growth rate of allometric growth scale coefficient slowed down. From a spatial point of view, the level of economic development in Northeast China is higher than that of electricity consumption, while the level of economic development in the central and western regions is lower than that of electricity consumption, indicating that the degree of energy intensive use in Northeast China is lower than that in the central and western regions. Therefore, with the progress of economic level and the continuous innovation of science and technology, the development of new energy has a significant impact on allometric growth. New energy has a good development prospect for optimizing the power structure and promoting the transformation of the power industry.

4.2 countermeasures and suggestions

(1) Adjust power source structure.

Since entering the new normal, China's economic growth rate has continued to slow down with double-digit high growth, and the economic growth rate has decreased, so that the growth rate of electricity demand has also decreased. With the increasingly prominent environmental problems, China began to implement energy conservation and emission reduction policies in 2007. On the one hand, economic development needs the support of electricity. On the other hand, China's power industry is also a key area of pollutant emission reduction. Therefore, the power industry should gradually shift from meeting the ' quantity ' demand to focusing on meeting the ' quality ' demand. China ' s electric power production structure is quite different from that of other developed countries. Power production is dominated by thermal power. In 2017, thermal power accounted for 71.8% of all power generation, hydropower accounted for 18.3%, and non-water renewable energy generation accounted for only 6.0%. To solve the environmental bottleneck faced by China's power development and provide high-quality power energy for economic development, it is necessary to optimize the power supply structure. Changing the dominant position of non-clean energy in the energy structure and gradually increasing the proportion of clean energy can not only meet the demand of economic development for electricity, but also reduce the environmental pollution caused by thermal power coal, so as to achieve the purpose of emission reduction.

(2) Optimize power layout.

China's eastern, central and western regions uneven distribution of energy resources. The power generation resources in the eastern and central regions are relatively scarce, but the power load is large. The centralized construction of a large number of coal-fired power plants has caused problems such as environmental capacity shortage in the eastern and central regions, excessive dependence on coal transportation for energy transportation, and the phenomenon of power shortage and coal shortage has repeatedly occurred. On the contrary, the western region is rich in power generation resources such as coal, hydropower and wind energy. However, due to the lagging economic development, the electricity load is relatively small and the electricity supply is relatively surplus. In recent years, clean energy such as wind power and photoelectricity has been greatly developed in Gansu, Xinjiang and other places. However, due to the weak ability of local power grid to accept wind power, the limited local consumption ability and the insufficient transmission channel, the abandonment of wind and light is serious. Therefore, optimizing the distribution of power, speeding up the construction of power grid, through the introduction of clean energy replacement and transmission to outside the area of consumption, can achieve the effect of optimal allocation of resources and macro energy saving and emission reduction at the national level or in a large range, that is, to achieve the goal of reducing energy consumption in the transmission area and to achieve the purpose of pollution reduction in the receiving area. At the same time, wind power transmission can transform the resource advantages of Northwest China into economic advantages, alleviate the pressure of supply and demand and environmental protection in the Middle East, which is the objective requirement of implementing the national energy strategy and promoting the economic development of Northwest China.

(3) Formulate differentiated regional economic development policies.

At present, there is still a large gap in the level and mode of economic development between the eastern, central and western parts of China, which also leads to significant differences in the relationship between electricity consumption and abnormal growth of regional GDP. Therefore, in the formulation of regional development policies, regional differences should be fully considered and adapted to local conditions. For the eastern region, we should develop in transition, attach importance to innovation-driven, develop new economic growth models such as sharing economy and platform economy, and reduce the pressure on resources, environment and electricity. For the western region, taking into account the obvious differences in power consumption intensity between the eastern region and the western region, there is a large space to improve the efficiency of power input. It is necessary to transform in development, pay attention to technological progress and improve the efficiency of resource allocation. On the one hand, we should fundamentally change the current economic development model, curb the blind investment and low-level redundant construction of traditional industries with high energy consumption and high pollution, actively promote the development of emerging service industries and advanced manufacturing industries based on the direction of industrial structure diversification, gradually reduce the proportion of the secondary industry in the three industries, and gradually increase the proportion of the tertiary industry in the industrial composition, thereby reducing the energy consumption of economic development. On the other hand, China should strengthen the research and development of energy storage technology and formulate corresponding measures to vigorously promote the use of clean energy to ensure that the western region can obtain higher income from power output and realize the fairness of the impact of power development on regional economy. (line510-600)

Recommendation 7 : Compared with the existing literature, what are the theoretical contributions and application values of this study ? It is suggested to strengthen the corresponding discussion in the conclusion part. 

Response : Thank you for your valuable comments. According to your opinions, we add the theoretical contribution and application value of this paper at the end of the research conclusion. The specific modifications are as follows :

Based on the previous research results, this paper innovatively introduces the allometric growth model in biology, broadens the application scope of the method, and realizes the interdisciplinary application of the method. From the perspective of multi-scale and multi-level, this paper analyzes the relationship between electricity consumption and allometric economic growth, which can better provide theoretical support for the rational use of regional electricity in China, provide scientific basis for the formulation of differentiated regional electricity regulation policies, provide application value for cross-section empirical research, and promote the coordinated development of national economy and electricity production. (line645-654)

Recommendation 8 : English expression needs to be more precise. 

Response : Thank you for your valuable comments on our manuscript. We have carefully revised and polished the manuscript according to the opinions. And the expression in the manuscript was improved.

Recommendation 9: It is recommended that the author increase the reasons for choosing China as a case study. 

Response : Thank you for your valuable comments on our manuscript. Your recommended literature is very helpful to us, we have carefully read and cited. Our specific modifications to this part are as follows ( Line ) :

China has a vast territory, and there are obvious spatial differences in natural resource endowment, economic development foundation and industrial structure characteristics. The utilization rate of resources varies greatly, and the potential to meet the needs of economic development is different[32]. Because of the ' West-Central-East ' gradient model, the eastern provinces of China have better spatial, economic and technological advantages than the western provinces[33]. From the perspective of different development stages, the intrinsic dependence between electricity consumption and economic growth is different due to energy policies and economic policies[21,22]. Therefore, the complex relationship between electricity consumption and economic development shows obvious scale effect. Therefore, understanding the characteristics of different regions can effectively integrate and allocate resources such as talents, funds, materials, and information, thereby accelerating the coordinated economic development of underdeveloped regions[34]. In addition, China 's clean energy resources are abundant, and in recent years, China 's clean and low-carbon process has been accelerating. Many indicators such as hydropower, wind power, photovoltaic, and nuclear power installed capacity under construction have remained the world 's first, and the world 's largest clean power generation system has been built. It has become an important force to promote the development of global clean energy. At this stage, the advantages of energy consumption mode change are huge, and it will continue to develop rapidly in the future. Based on this, this paper introduces the allometric growth model and combines the spatial econometric analysis method to analyze the spatial correlation characteristics of regional GDP electricity intensity. (line117-139)

Thank you again for your valuable amendments to improve the quality of research, and hope that if you feel that there are still shortcomings, please point out that we will continue to improve.

---

## [Decision Letter · Decision Letter 1]

15 Aug 2023

PONE-D-23-00464R1The Allometric Growth Relationship Between Electricity Consumption and Economic： A case of ChinaPLOS ONE

Dear Dr. Qi,

Thank you for submitting your manuscript to PLOS ONE. After careful consideration, we feel that it has merit but does not fully meet PLOS ONE’s publication criteria as it currently stands. Therefore, we invite you to submit a revised version of the manuscript that addresses the points raised during the review process.

We look forward to receiving your revised manuscript.

Kind regards,

Xiaobao Yu

Academic Editor

PLOS ONE

Journal Requirements:

Reviewers' comments:

Reviewer's Responses to Questions

**Comments to the Author**

1. If the authors have adequately addressed your comments raised in a previous round of review and you feel that this manuscript is now acceptable for publication, you may indicate that here to bypass the “Comments to the Author” section, enter your conflict of interest statement in the “Confidential to Editor” section, and submit your "Accept" recommendation.

Reviewer #1: All comments have been addressed

Reviewer #2: All comments have been addressed

2. Is the manuscript technically sound, and do the data support the conclusions?

Reviewer #1: Yes

Reviewer #2: Yes

3. Has the statistical analysis been performed appropriately and rigorously? 

Reviewer #1: Yes

Reviewer #2: Yes

4. Have the authors made all data underlying the findings in their manuscript fully available?

Reviewer #1: Yes

Reviewer #2: (No Response)

5. Is the manuscript presented in an intelligible fashion and written in standard English?

Reviewer #1: Yes

Reviewer #2: Yes

6. Review Comments to the Author

Reviewer #1: All comments have been addressed, but some minor problems still exist as follows:

(1) Syntax error and title consistency issue on line11-13: --study the relationship between electricity consumption and allometric economic growth in China at multiple scales..--

(2) There is one blank cell should be filled on top line in Table2 .

(3) In Fig.4, different colors stand for conomic zones, but it is not clear in the figure.

So, authors need to spend more time to revise the manuscript carefully again.

Reviewer #2: The authors have improved the paper according to the reviewers' comments, this paper can be accepted after making the following minor revisions.

1. The title needs to be modified to: The Allometric Growth Relationship Between Electricity Consumption and Economics in China.

2. Some minor errors in the abstract need to be corrected.

3. The introduction of methods should precede the introduction of data.

4. The title of the third section should be Results, and the sub-headings of the third section are too long and should be more concise.

5. The title of the fourth part should be Discussion, the authors should focus on describing the differences between the article study and other scholars' studies, thus highlighting the relevance and academic value of the article. This literature should be helpful for your research: Compilation of Water Resource Balance Sheets under Unified Accounting of Water Quantity and Quality, a Case Study of Hubei Province.

6. The conclusion part of the paper should give a brief introduction of the research background and methods, and then introduce the main conclusions.

7. The format of the full text and references should be standardized.

7. PLOS authors have the option to publish the peer review history of their article (what does this mean?). If published, this will include your full peer review and any attached files.

Reviewer #1: No

Reviewer #2: No

---

## [Author Response · Author response to Decision Letter 1]

24 Aug 2023

Dear reviewer and editor,

We are very grateful for your affirmation of our manuscript and your valuable comments to us. These comments are very valuable and helpful to the revision and improvement of our manuscript. We have carefully studied your comments and completed the correction. We hope it will be approved. The main corrections in the paper and responses to your comments are as follows:

Reviewer #1: All comments have been addressed, but some minor problems still exist as follows:

Point 1: Syntax error and title consistency issue on line11-13: --study the relationship between electricity consumption and allometric economic growth in China at multiple scales..--

Response 1: Thank you for your suggestion. According to your suggestion, we have made the following modification:

line 12-13 --study the allometric growth relationship between electricity consumption and economics in China at multiple scales.--

Point 2: There is one blank cell should be filled on top line in Table2 .

Response 2: Thank you very much for your suggestion. According to your suggestion, we have filled in the blank in Table 2: 

line 387-388 Table 2. Scale coefficients of electricity consumption and GDP abnormal growth at different stagesof economic development in the eight economic zones

Regions 1972-1978 1979-2003 2004-2017

Northeast region 1.0215 2.7375 2.1534

Northern coast 0.7148 2.128 1.5209

Eastern coast 0.8614 1.918 1.5649

Southern coast 0.3877 1.5045 1.6328

Middle reaches of the Yellow River 0.5685 1.9769 1.5343

Middle reaches of Yangtze River 0.5356 2.1687 1.67

Southwest region 0.8229 1.7864 1.6132

Northwest region 0.5839 1.8207 1.0687

Point 3: In Fig.4, different colors stand for conomic zones, but it is not clear in the figure.

Response 3: Thank you for pointing out the issue about the figure. According to your suggestion, we have made the following modification:

line 354-355 

Fig.4 Relationship between electricity consumption and GDP growth in the three economic zones

Reviewer #2: The authors have improved the paper according to the reviewers' comments, this paper can be accepted after making the following minor revisions.

Point 1: The title needs to be modified to: The Allometric Growth Relationship Between Electricity Consumption and Economics in China.

Response 1: Thank you for your suggestion. According to your suggestion, we have revised the title to: The Allometric Growth Relationship Between Electricity Consumption and Economics in China.

Point 2: Some minor errors in the abstract need to be corrected.

Response 2: Thank you for pointing out the error in my article. We have checked and revised the content of the abstract, and the revised content is as follows:

line 6-31 Abstract: Electric power is the basic industry of a modern country. The rapid development of the power industry has promoted economic development and social progress. With the establishment of the carbon neutrality target of carbon peak, China’s power industry is also facing new situations and new challenges. This paper innovatively introduces the concept of allometric growth in biology, and uses the data of provincial electricity consumption and economic development from 1972 to 2017, to study the allometric growth relationship between electricity consumption and economics in China at multiple scales. Combined with the spatial autocorrelation analysis method, the temporal and spatial characteristics of allometric growth between electricity consumption and economic development were researched based on the analysis of spatial correlation of GDP electricity consumption intensity. The results show that: (1) The provincial GDP electricity consumption intensity shows an aggregate distribution, and the local aggregate pattern changed significantly from 1972 to 2017. (2) The High-High Cluster gradually disappeared, and the Low-Low Cluster moved southward, appearing in the middle and lower reaches of the Yangtze River. (3) The differential growth coefficient of electricity consumption and economic development increased initially and decreased from 1972 to 2017. (4) GDP would increase by 7.4%, 20.1%, and 15.6% in the simulation of electricity consumption, increasing 10% at the three stages of China's economic development. (5) The allometric growth coefficients of electricity consumption and economic development are very different in the three economic belts, eight economic zones, and provinces with different characteristics at the different stages of economic development. Formulating differentiated regulation policies for regional economic development and electricity consumption will be conducive to the coordinated development of electric power production and the national economy.

Point 3: The introduction of methods should precede the introduction of data.

Response 3: Thank you for your suggestion. According to your suggestion, we have adjusted the order in which methods and data are introduced.

Point 4: The title of the third section should be Results, and the sub-headings of the third section are too long and should be more concise.

Response 4: Thank you very much for your valuable comments. We have changed the title of the third section to Results, and simplified some of the sub-headings in the third section.

line 326-328 3.2 The Allometric Growth Relationship Between Electricity Consumption and Economics 

3.2.1 China

line 356 3.2.2 Three Economic Belts

line 390 3.2.3 Eight Economic Zones

line 419 3.2.4 Province (City)

Point 5: The title of the fourth part should be Discussion, the authors should focus on describing the differences between the article study and other scholars' studies, thus highlighting the relevance and academic value of the article. This literature should be helpful for your research: Compilation of Water Resource Balance Sheets under Unified Accounting of Water Quantity and Quality, a Case Study of Hubei Province.

Response 5: Thank you for your suggestion. According to your suggestion, we have changed the title of the fourth section to Discussion. In this section,we have quoted the article you recommended and described the differences between the article study and other scholars' studies. The revised content is as follows:

line 452 4. Discussion

line 463-467 In this paper, the allometric growth relationship between electricity consumption and economics in China has been studied at multiple scales. In contrast, some articles from the perspective of water resources improved the value accounting method for water resources by assessing water efficiency and quality [37].

line 723-726 37. Liang Yuan; Liwen Ding; Weijun He; Yang Kong; Thomas Stephen Ramsey; Dagmawi Mulugeta Degefu; Xia Wu. Compilation of Water Resource Balance Sheets under Unified Accounting of Water Quantity and Quality, a Case Study of Hubei Province[J]. Water,2023,Vol.15(1383): 1383

Point 6: The conclusion part of the paper should give a brief introduction of the research background and methods, and then introduce the main conclusions.

Response 6: Thank you very much for your valuable comments. A brief introduction of the research background and methods has been included in the concluding section of the paper.

line 552-557 Electricity development is closely linked to economic and social development. In this article, combined with the spatial autocorrelation analysis method, the temporal and spatial characteristics of allometric growth between electricity consumption and economic development were researched based on the analysis of the spatial correlation of GDP electricity consumption intensity. The conclusions are as follows:

Point 7: The format of the full text and references should be standardized.

Response 7: Thank you for your suggestion. We have adjusted the format of the full text and references.

line 28 the different stages of economic development. Formulating differentiated regulation

line 46 and people's life [3]

line 47 supply [4]

line 636-637 6. Yoo S H. Electricity consumption and economic growth: evidence from Korea[J]. Energy Policy, 2006, 1(1/4):235-243.

...

Yi Qi, Mengyuan Tang, Dongsheng Dang and Caijuan Qi

Aug. 24, 2023

---

## [Decision Letter · Decision Letter 2]

4 Sep 2023

The Allometric Growth Relationship Between Electricity Consumption and Economics in China

PONE-D-23-00464R2

Dear Dr. Qi,

We’re pleased to inform you that your manuscript has been judged scientifically suitable for publication and will be formally accepted for publication once it meets all outstanding technical requirements.

Kind regards,

Xiaobao Yu

Academic Editor

PLOS ONE

Reviewers' comments:

Reviewer's Responses to Questions

**Comments to the Author**

1. If the authors have adequately addressed your comments raised in a previous round of review and you feel that this manuscript is now acceptable for publication, you may indicate that here to bypass the “Comments to the Author” section, enter your conflict of interest statement in the “Confidential to Editor” section, and submit your "Accept" recommendation.

Reviewer #1: All comments have been addressed

Reviewer #2: All comments have been addressed

2. Is the manuscript technically sound, and do the data support the conclusions?

Reviewer #1: Yes

Reviewer #2: Yes

3. Has the statistical analysis been performed appropriately and rigorously? 

Reviewer #1: Yes

Reviewer #2: Yes

4. Have the authors made all data underlying the findings in their manuscript fully available?

Reviewer #1: Yes

Reviewer #2: Yes

5. Is the manuscript presented in an intelligible fashion and written in standard English?

Reviewer #1: Yes

Reviewer #2: Yes

6. Review Comments to the Author

Reviewer #1: The manuscript had been modified better and better while some minor errors should be corrected for using uppercase on first letter of first word after some serial numbers like "3.1.3 local autocorrelation analysis".

Reviewer #2: In the revised manuscript, you have provided a detailed response to my review comments and improved the manuscript. The quality of the paper has been significantly enhanced. I think your work has reached the level suitable for publication, and I recommend it for publication.

7. PLOS authors have the option to publish the peer review history of their article (what does this mean?). If published, this will include your full peer review and any attached files.

Reviewer #1: No

Reviewer #2: No

---

## [Editor Report · Acceptance letter]

12 Sep 2023

PONE-D-23-00464R2 

The Allometric Growth Relationship Between Electricity Consumption and Economics in China 

Dear Dr. Qi:

I'm pleased to inform you that your manuscript has been deemed suitable for publication in PLOS ONE. Congratulations! Your manuscript is now with our production department. 

Kind regards, 

on behalf of

Dr. Xiaobao Yu 

Academic Editor

PLOS ONE